# Evaluation of Online Counseling through the Working Experiences of Mental Health Therapists Amidst the COVID-19 Pandemic

**DOI:** 10.3390/healthcare12040495

**Published:** 2024-02-19

**Authors:** Maria Moudatsou, Areti Stavropoulou, Michael Rovithis, Sofia Koukouli

**Affiliations:** 1Department of Social Work, Faculty of Health Sciences, Hellenic Mediterranean University, 71410 Heraklion, Greece; koukouli@hmu.gr; 2Laboratory of Interdisciplinary Approaches for the Enhancement of Quality of Life (QoLab), 71410 Heraklion, Greece; astavropoulou@uniwa.gr (A.S.);; 3Department of Nursing, Faculty of Health and Care Sciences, University of West Attica, 12243 Athens, Greece; 4Faculty of Health, Science, Social Care and Education, Kingston University, London KT2 7LB, UK; 5Department of Business Administration and Tourism, School of Management and Economics Sciences, Hellenic Mediterranean University, 71410 Heraklion, Greece

**Keywords:** online counseling, mental health professionals, COVID-19 pandemic, internet psychotherapy, qualitative study, Greece

## Abstract

This study aimed to reflect on mental health professionals’ experiences with online counseling during the COVID-19 pandemic, as well as their perceptions and recommendations for the future. The method of qualitative research with semi-structured interviews was used. The sample consisted of 17 mental health professionals working in the public or private sectors. A framework analysis revealed four main themes, namely (a) the evaluation of online counseling; (b) comparing in-person and online counseling; (c) factors influencing the effectiveness of online counseling; and (d) suggestions for the future use of online counseling. Most therapists reported that their overall experience with online counseling was positive. The main advantages cited were the accessibility for everyone and the reductions in time, money, and distance. Its primary drawbacks included less nonverbal communication, the inability to employ certain therapeutic tools, problems with confidentiality, lack of experience, and technical difficulties during online sessions. Its effectiveness depends on contextual factors and factors related to the therapeutic process itself. Organizational planning, training, and a solid implementation strategy may help ensure that this communication medium is used to its fullest potential. In addition, the possible utilization of remote counseling combined with in-person psychotherapeutic intervention methods will provide solutions for the future, especially in crisis situations.

## 1. Introduction

Prior to the COVID-19 pandemic, counseling and psychotherapy were primarily conducted in-person. Even though remote counseling has been used in the past, there have been few reasons to justify its integration into everyday practice, and mental health professionals were particularly cautious [1,2].

The pandemic of COVID-19 caused great tension and anxiety to people worldwide, influencing their mental health [3]. Even after the end of the first wave, stress remained due to the pandemic’s multiple ramifications (such as isolation, social distancing, job loss, and fear of contracting the disease), leading to a secondary epidemic related to negative effects on mental health [4,5]. Remote intervention became necessary because of the pandemic in order to continue the treatments of those who needed them, while also reducing the spread of the virus [6].

Online counseling was first introduced in the 1960s in order to meet the needs of clients who were isolated in mountainous or remote areas and had to travel a long distance to treatment centers, or to save time and money [7]. However, before the COVID-19 pandemic, counseling and psychotherapy were generally provided in-person. Therefore, most therapists did not have prior knowledge of or experience with online counseling and were quite hesitant to use it, considering that person-to-person intervention is more effective [8]. Nevertheless, due to the challenging circumstances of COVID-19, they were required to initiate or continue their treatments primarily online, without any previous preparation or a well-defined treatment protocol [9].

The rapid changes in health and society have indorsed the use of emerging technologies in many areas of healthcare. More specifically, in the field of counseling, with issues related to immediate accessibility, privacy, and effectiveness, and where the maintenance of communication between therapists and clients in a period in which social distancing was necessary, the use of online practices proved to be a substantial action for the continuation of the therapeutic process [7,8,10,11]. The effectiveness of these practices, however, lies in understanding and being able to manage both the advantages and disadvantages of these technologies, such as security, confidentiality, and the lack of a code of ethics [7,8]. Furthermore, issues of accessibility due to economic, social, and contextual factors should be identified for creating effective platforms for both therapists and clients [9].

In the past, several therapists have successfully utilized online counseling, arguing that it does not differ from in-person therapy [10]. However, in many countries, such as France and Belgium, online counseling was used at least in the first lockdown, without previous experience or training of the therapists, raising concerns about its effectiveness and, more importantly, compliance with counseling ethics [11]. Some therapists still have reservations about ethical and privacy issues, as they believe that ethics are not sufficiently respected [7,8].

Online counseling offers a number of benefits. It can provide opportunities and perspectives like easy access to therapy in case of geographically remote places, or in circumstances of illness or relocation to another area, and resolve issues like reducing distance, saving money, and, in general, making treatment more flexible [9,12].

Despite the advantages of online counseling, there is difficulty for therapists and clients to develop a therapeutic relationship as they cannot sufficiently connect and empathize [13]. Some have mentioned difficulties with the technology and internet connectivity. Others continue to be wary about ethical and privacy issues, considering that ethics are not sufficiently observed [7,8].

The therapist’s knowledge, training, practice, work experience, and expertise are central to ensuring appropriate professional ethics [14]. The effectiveness of internet counseling might also be influenced by the therapist’s psychotherapeutic approach. It has been found that cognitive therapists are more familiar with it than psychodynamic therapists [15].

Online counseling does not provide the therapist the possibility of global and comprehensive monitoring of the client. As a result, valuable information is lost from non-verbal communication [16]. Additionally, it is not always appropriate for crisis situations, such as when a client expresses suicidal or self-harming intent, or when therapists feel uncomfortable with it [17,18].

Even though it is considered necessary and beneficial, online counseling training is only occasionally offered, at least in European countries [19,20]. The American Association of Psychotherapists’ recommendations for online therapy suggest that therapists should feel confident utilizing the technology themselves and have ensured that their clients are also familiar with its use [11]. Additionally, it is essential to establish a secure framework and a comprehensive work protocol that will define the guidelines of online counseling. It is required to be implemented in the context of a contract regulating detailed issues of ethics, methodology, and terminology [11].

As already mentioned, training is necessary for therapists and everyone involved in therapy. The training and specialization should be accompanied by their formal certification following all the rules of ethics and ethics of counseling, while also respecting the client’s needs and confidentiality [19,21]. Support from the relevant health agencies is needed in addition to the political, economic, and scientific input from everyone involved in the development of health policies [22].

Remote counseling exhibited several advantages, including the following: (1) it supported people being treated for both their pre-existing and pandemic-related mental health issues, and (2) it helped reduce the transmission of COVID-19. However, it is a topic that has not been thoroughly investigated, especially during the COVID-19 pandemic. Moreover, the endorsement of online counseling in Greece has not been explored adequately, and there is a need to fill in this knowledge gap by providing reliable research evidence. It is important to note that, especially in the public sector, online counseling was endorsed with a considerable delay, as the system was not ready for this abrupt change. In contrast, online procedures were applied faster in the private sector. Not many studies have been conducted to investigate the views and experiences of health professionals on using online counseling under these circumstances. As such, producing qualitative research evidence in this area was considered essential for answering the topic under investigation in depth. This research approach aims to better understand the circumstances surrounding online counseling as a selection of therapy, to evaluate the concerns of experts, the problems identified, and any potential benefits. The present study aimed to investigate the aforementioned issues.

## 2. Methods

### 2.1. Aim

This study examined the work contexts, perspectives, and beliefs of mental health practitioners about remote therapy. In particular, their overall experience with online therapy (phone, internet, skype, viber, messenger, etc.) was investigated, as well as the benefits and main obstacles that remote therapy presented to the practitioners, clients, and families. Additionally, recommendations for the usage of online therapy in the future were explored, along with ethical issues.

### 2.2. Study Design

A qualitative study approach was favored to elucidate the essence of lived experience and develop composite descriptions. Qualitative research investigates the perceptions, feelings, and experiences of the subjects under study in depth through a holistic approach [23,24] As the main goal of this research was to thoroughly examine the subjective experiences of health professionals regarding online counseling, the implementation of a qualitative research design was considered the most appropriate one.

### 2.3. Population and Study Sample

Mental health professionals, mainly psychologists, and social workers, working either in the public or private sectors constituted the study population. The participants of the sample were selected using a purposeful sampling technique. This non-random sampling technique is also known as judgmental sampling, as the researcher depends on their own judgement for selecting those individuals who are more eligible to participate in a study. According to this technique, the researcher identifies, from the population under investigation, those members who meet the study criteria and can adequately and deeply address the research questions. This means that the selected study participants are extremely knowledgeable about the topic under investigation and can provide a wealth of information about it [24,25]. The inclusion criteria for participating in the study was to practice counseling for at least 2 years and have experience with practicing online counseling during the pandemic period. This experience was considered essential for discussing their experience on online counseling in depth. Specialization in a psychotherapy approach was also considered desirable. The mental health professionals selected were practicing remote counseling either in public agencies or in the private sector. Most of the participants were practicing in the private sector, as the online counseling services in the public sector were not implemented immediately. Our study was advertised through social media (FB and Instagram). Professionals who were therapists were contacted by one member of our authoring team to explore their interest in participating in our study. Access to our sample was also achieved by contacting these professionals directly at their workplaces, or through their professional associations. To obtain the participants’ consent, detailed information regarding the nature and the goals of the study as well as the inclusion criteria were initially provided by phone and then via email. The voluntary nature of participation in the survey and the confidentiality of their responses were also assured. The recruitment process lasted approximately 2 months. Twenty-five (25) mental health professionals from four (4) metropolitan cities in northern, central (Attica region), and southern (Grete region) Greece, agreed initially to participate in our study. Eight (8) of them were not finally involved in the study due to time restrictions and lack of availability. Sample size was determined through a data saturation process. Data saturation occurred after completing the 17th interview. At this point, the new data that emerged did not offer new information to allow for further coding and categorization or to seek for additional participants.

### 2.4. Sample’s Socio-Demographic Profile

Seventeen (17) mental health practitioners constituted the final sample, specifically two (2) mental health counselors, five (5) social workers, and ten (10) psychologists. All participants were women and married with children, and most of them were over 50 (50–65 years old). Also, five mental health professionals were university graduates, eleven had a postgraduate degree, and one was a PhD holder. Most of the therapists said that they treat more than thirty clients every week, the majority of whom are adults. Regarding their psychotherapeutic approach, twelve implement a systemic approach, two use a psychodynamic approach, and one did not specify, while two had no specialization. Additionally, seven of them had between twenty and thirty years of experience as psychotherapists, six were public employees, and eleven had a private practice (see Appendix A).

### 2.5. Data Collection

For the data collection, semi-structured interviews were used. Semi-structured interviews were used to explore the in-depth personal experiences, perceptions, and thoughts of the participants for the topic under investigation [26].

The interview scheme focused on two main axes: the subjective experience with online counseling and their online counselling challenges as these were confronted by the professionals involved. More specifically, the semi-structured interview guide comprised one introductory open-ended question asking the participants to describe their experience with online counseling. Furthermore, the interview guide focused on the positive and negative aspects of online and recommendations for future use. The relevant literature on online counseling during the period of COVID-19 was the main source of information used for the development of the main axes and the open-ended questions used in the interview guide.

The first author (M.M.), who is experienced in qualitative research approaches, methodically conducted online interviews with all respondents within a designated timeframe (May–June 2022). None of the selected participants withdrew from the study. The average length of an interview was 30 to 40 min.

### 2.6. Data Analysis

The interviews were first recorded and then transcribed by the first author. To analyze the data, the framework analysis was used [27,28]. This method is popular for managing and analyzing large amounts of qualitative data deriving usually from semi-structured interviews. It is used by multidisciplinary research teams to obtain a clearly descriptive and comprehensive outline of the research data. It leads to structured outcomes and supports the researchers in identifying themes and connotations that might emerge from the data. Framework analysis is valued for following a systematic management and data analysis approach, thus providing a clear audit trail throughout the research process that guarantees the production of reliable findings [24,28,29]. The utilization of framework analysis was considered as most appropriate for our study, as our team consisted of researchers from different scientific backgrounds, and the large amounts of data derived from the semi-structured interviews had to be managed in a highly structured manner.

The framework analysis includes the following stages: (a) familiarization with the data by the researchers as all the results were thoroughly studied; (b) the coding of the data; (c) the identification of the themes and categories that emerged; (d) recording the results based on the existing themes; and (e) writing the results [29].

The trustworthiness of the study was ensured by using mainly two techniques: investigator triangulation and reflexivity. The first technique involved two members of the research team who examined, coded, interpreted, and analyzed the survey data, while the second technique focused on examining how the researchers’ preconceptions and values might have influenced the decisions taken throughout the study process. Furthermore, member checking was used to ensure that the findings were representative of the participants’ experiences and that no important issues were missed. To accurately present the research data, a backward translation technique was used [23,24,25].

To report the study results, the COREQ guidelines were used. COREQ is a checklist with 32 items grouped into three domains addressing the (a) research team and reflexivity, (b) study design, and (c) data analysis and reporting. This checklist may assist qualitative researchers to report all the important dimensions of their research design and implementation as well as their findings in a robust and comprehensive manner [30]. In the present study, issues regarding the study design (e.g., sampling strategy and recruitment, sample characteristics, interview guide, and data saturation), the analysis and findings (e.g., theme generation, participants’ quotations), and the researchers’ profiles and involvement were presented following the COREQ guidelines to the extent possible.

The research team consisted of academics with a wide range of backgrounds in health and social care services (social work, social policy, and nursing), as well as expertise in qualitative research. All of them contributed equally to the design of the research and provided support at all stages of the study process.

### 2.7. Ethical Issues

Written permission was granted by the university’s Research Ethics Committee (A.A. 99, A.P. 65/18-11-2021, accessed on 18 November 2021). The managers of the private practices or the heads of the public agencies where the interviewees worked also granted their approval. Each participant provided her consent after being notified through email or social media (messaging) of the study’s purpose, its voluntary nature, and the confidentiality of the information collected.

## 3. Results

Four main themes were derived from the data analysis, describing the subjective working experience of the participants with online counseling. The four themes revealed are as follows: (1) the evaluation of online counseling (working experience with online counseling, overall assessment, and advantages/disadvantages); (2) comparing in-person with online therapy (similarities, differences); (3) factors influencing the effectiveness of online counseling (contextual factors and the therapeutic process); and (4) suggestions for its future use (a supplementary tool used with caution, create the appropriate conditions) (See Appendix A).

### 3.1. Evaluation of Online Counseling

#### 3.1.1. Working Experience with Online Counseling

Some therapists have already been using online counseling services prior to the COVID-19 period due to the specific conditions of their clients (living far away, work conditions, studies, etc.). However, almost all respondents began using online therapy more systematically during the COVID-19 period, albeit with hesitation.

“I started, experimentally a few years before COVID-19 when some of my clients decided to relocate to another city for various reasons”. (P11).

“Before COVID it (the online counseling) was possible to serve some cases outside of the county. Then, it was used solely throughout the pandemic and now things are somewhat back to normal”. (P8).

“I began using it during the first quarantine. Before, there was no need and I was very reluctant for various reasons...I had to evaluate how the basic principles of our work will be safeguarded…” (P5).

Some, even throughout the COVID-19 period, did not prefer to use it. They did so only in case of illness of themselves or their clients.

“During the COVID period I did face-to-face sessions. My clients didn’t want it. Only in situations where a person had received a personal or family diagnosis of COVID we used online therapy”. (P4).

#### 3.1.2. Overall Assessment

In the end, for the majority, the overall assessment was favorable. They evaluated online psychotherapy positively, despite the lack of prior experience and the cautiousness of the first transitional phase. They agreed that it was essentially an issue of familiarity of the parties involved, as well as of the therapist’s knowledge, experience, ability to adapt, and willingness to learn new things.

“(My experience was...) excellent. Online didn’t hinder me in any way. I put new members in groups, graduations took place online…. (all these were…) concrete signs that everything went well. It matters how one feels comfortable”. (P1).

“Now, I prefer it. I sense the closeness. As though we are face to face. There are many benefits and making modifications is simple and easy. I tried myself on something new. Change is never easy, but I was able to adapt”. (P10).

“At first, I took each step very cautiously. I believed in face-to-face counseling. Now, however, I am certain. My interventions were successful. It worked well in restrained circumstances. My clients are very satisfied”. (P13).

“Although I was afraid that it would be difficult to build a therapeutic relationship between me and the client, eventually it was effective. My experience is very positive”. (P11).

#### 3.1.3. Advantages and Disadvantages of Online Counseling

The main advantage mentioned was rendering counseling services more flexible to the therapist and the client. The majority stressed that it reduces travel distance and time and ensures the accessibility of socially excluded people or those who would not attend if the sessions were face-to-face.

“You can work online from any location. The primary benefit is this. The treatment can be continued from anywhere at any time...For people with disabilities and mobility problems, it is very helpful. Also, some of our young clients are worried that their fellow classmates might see them, and, for that reason, they feel more exposed. In that case, video conferences were helpful”. (P5).

“It is a valuable supplementary tool. It reduces the physical distance that would make a live session prohibitive. It also helped the clients in teamwork”. (P3).

“It gave an opportunity to elderly family members who are not accustomed to technology or therapy to become acquainted with both. Furthermore, after one family member began using it, the others got to know it as well. For instance, a mother we had to call to participate might show up for online sessions, but not for live ones. Through the online program, participants could observe, either directly or indirectly, how the therapist interacts with clients”. (P10).

Nevertheless, they enumerated many drawbacks. A primary disadvantage is the difficulty of assessing and using nonverbal communication and the risk of poorer quality of interaction due to contextual factors.

“The internet gives the opportunity to focus on the gaze, but the rest of the body is vanished. The transference and countertransference required, such small qualities are lost…. In face-to-face communication feelings and unconscious experiences are shared differently…”. (P8).

“It’s less interaction. It is not comprehensive……you don’t have the overall image of the person in front of you...there are other distractions during sessions (the phone etc) In other words, there are other stimuli besides counseling and psychotherapy”. (P15).

Certain participants mentioned that, in online counseling, they do not have the possibility to use all the therapeutic tools.

“You can’t utilize all of your tools, like music or painting. Nonverbal cues are invisible to you; you see them, but not very much”. (P7).

Additionally, some raised ethical issues, while for others, there was no question of violating ethics in either context.

“… the professionals suspect that someone is recording them or someone else is in the room. The client is similarly concerned about this…”. (P5).

“Also, a difficulty reported by clients was maintaining privacy from their own space. They said that they do not feel safe when people in the next room or apartment are known and can hear. When the client does not feel safe, this limits the session and its benefits”. (P9).

A number of professionals view distance as a barrier to establishing a therapeutic relationship. Conversely, others think that developing a therapeutic relationship is unaffected by distance. The client and the therapist jointly determine the extent to which the client wants to engage in therapy.

“I find it facilitates superficiality. There is no particular connection. A more superficial therapeutic relationship”. (P17).

“Maybe it takes more time to build a good therapeutic relationship. The information you lose in online you gain along the way. More sessions might be required”. (P5).

“In my experience, establishing a therapeutic relationship is not a problem. A person will work on his/her problems if he/she wants to. Additionally, the therapist will grow and acquire new abilities. If it is his desire... For both the therapist and the person treated, it is an issue of free will and decision”. (P2).

Certain advantages, such as staying in your own place, may also become drawbacks affecting the quality of the meetings. This is also connected to the lack of a treatment protocol.

“While we can wear comfortable clothing in sessions, previous circumstances taught us the importance of self-care and professional dressing. Personally, I think this is rather negative”. (P1).

“In my opinion, both the professional and the person being treated ought to establish clearer boundaries.”. (P9).

“People still lack the education necessary for distant learning; they attend online meetings in robes, in the car, at their parents’ house, and are not in the controlled setting of a therapist’s office where everything is turned off and therapy is the main focus”. (P4).

It was also emphasized that online counseling was ineffective for patients with certain conditions.

“Behavioral activation in those suffering from anxiety, social anxiety, or depression is not improved by it. It is not possible to implement therapies for children with behavioral issues, ADHD, or Asperger’s syndrome. Some child-centered interventions, such as play therapy, cannot be conducted remotely, unless a protocol is established. I believe these obstacles are difficult to overcome”. (P13).

One of the main issues raised was clients’ and therapists’ unfamiliarity with technology, which tends to immediately exclude those who are less knowledgeable. Concerns were also expressed about whether it is financially feasible for vulnerable social groups to acquire the necessary equipment.

“Not everyone is tech savvy. Age is significant, in my view. The challenges for 60 years old are different from 15 years old. Even for electronic payments”. (P14).

“A student eventually stopped coming to class, and I found out that she didn’t have a strong signal at home”. (P5).

### 3.2. Comparing In-Person and Online Counseling

#### 3.2.1. Similarities

Regarding the similarities, most respondents focused on the establishment of a therapeutic relationship, its goals and ethical principles, the interview and the verbal and—to a certain extent—the non-verbal communication.

“The goals and fundamental ethical principles, or the interview procedure, for instance, were the same. The interview’s format remained unchanged. It took the same amount of time. Case management and referrals were interchangeable”. (P5).

“Online privacy is equivalent to that of an office setting. Issues of ethics are similar”. (P4).

“I’m still attempting to establish a relationship with my client. Moreover, the therapeutic alliance endures”. (P7).

“In both situations, a relationship with certain qualities develops. There’s the non-verbal (limited to the face) and the verbal communication. As long as there is a contract, there is a result”. (P8).

#### 3.2.2. Differences

The context in which counseling is provided, the non-verbal communication, the whole procedure, and confidentiality issues are the primary areas of distinction.

“There is a screen in online counseling. The therapist does not see the client in its entirety. Inevitably, a loss of information occurs”. (P12).

“Live communication requires the person to leave their home and go to the therapist’s office, which is a neutral setting. This transition also functions similarly to a commitment process”. (P8).

“In live communication you are more focused; when you’re online, distractions like the phone and door ringing can happen. When we speak face-to-face and the room is empty, I feel more secure…..Confidentiality and privacy are important to me”. (P7).

### 3.3. Factors Influencing the Effectiveness of Online Counseling

#### 3.3.1. Contextual Factors

A number of therapists who work in the public sector, in particular, stated that the organization’s role is pivotal. Furthermore, it is essential to establish a contract and adhere to procedure.

“The agency’s organizational structure is crucial. A communication framework should be established, indicating the platform to be utilized as well. The client must have an informal contract”. (P5).

“I believe that therapists need to be trained and that platforms should provide protocols so that they know what cases to take on. For example, if we are communicating remotely with a woman who is experiencing domestic abuse, we need to make sure that she is protected. The same goes for suicidal clients; platforms should provide protocols that are followed. It is vital to preserve quality”. (P13).

“An initial framework is necessary…establish guidelines for cooperation. The rules that should be followed...should be outlined in a contract that is agreed at the outset”. (P4).

Participants also denoted technical concerns like expertise and network performance.

“I’ve experienced how having a poor network can make our work more challenging”. (P12).

“The network’s quality, available technology, tablets, laptops, and cell phones. This could lead to issues. There might be no signal on the cell phone. That creates a problem”. (P4).

#### 3.3.2. The Therapeutic Process

For internet counseling to be successful, the therapist’s own role is crucial, as well as their expertise, abilities, and background.

“The therapist is important. His/her expertise, experience, disposition, and engagement in the role. The conviction that it’s also possible to do it that way. We learn when we are motivated to achieve something. Zoom was a new experience for me. I discovered it”. (P1).

A number of participants considered that the therapeutic approach has an impact on the effectiveness of online counseling.

“Systemic therapy adherents are more flexible. Thus, the last group to deal with internet was psychoanalysts. It’s not the same for us systemics. We consider distance a useful tool. We are more detached”. (P1).

“I’m not sure if it would be effective when applied to psychoanalysis’s traditional dimension. Or for those who are body-centered and find it difficult to intervene. As a systemic, I don’t find it problematic. In the end, I believe the therapist’s method matters”. (P7).

However, other participants gave more priority to the therapeutic needs of the person treated.

“I think it has more to do with the therapist and the person’s needs rather than the approach used…. if the online process helps you achieve your goal. I work with people who have difficulties in socializing and interacting with others. You realize that in this case it is not in their favor to communicate online.....they didn’t improve”. (P9).

Almost all respondents agreed that the experiences of the clients positively or negatively influence the course of online counseling. Some had a positive experience, while for others, the experience was negative.

“They found it interesting. It didn’t seem difficult for them. I received positive reviews. Perhaps the assessments would differ if the two (online vs. in-person communication) were compared during a typical period (without COVID). They formed a social link, felt very supported, and experienced a sense of belonging to a team…” (P8).

“Many regular clients ceased coming during COVID-19. They did not feel comfortable. Maybe other persons were present in the room during sessions or lacked the resources”. (P5).

### 3.4. Suggestions for the Future Use of Online Counseling

#### 3.4.1. A Supplementary Tool Used with Caution

There are benefits to online counseling, and it can be used in the future where appropriate with some restrictions and by implementing a specific protocol respected by the client. However, almost all respondents agreed that one should use it to supplement in-person counseling rather than to replace it.

“Getting familiar with online will facilitate but not replace in-person…” (P1).

“We have to be careful about tools and confidentiality. For example, I tell them to wear headphones…. I ask them to confirm that they are alone in the room. They are not allowed to use drugs, alcohol, or cigarettes. When they attend the office and when they are online, the indicators will look different for example when a person communicates suicidal thoughts or thoughts of self-destruction. Additionally, the expert will treat you differently online. We must be prepared”. (P12).

“A protocol for the management and implementation of the teleconference ought to exist. Anticipate difficulties. Should Skype malfunction, you have the option to utilize an alternative platform”. (P5).

It can be utilized by those who are unable to participate in face-to-face therapy for a variety of reasons, such as health issues.

“It is just one additional tool available to therapists…. for those in hospitals, institutions, or with serious illnesses…and with a low cost”. (P6).

“Only when the person concerned cannot move, is sick, disabled, or far away”. (P17).

#### 3.4.2. Create the Appropriate Conditions

The education and training of therapists in online counseling is essential. Some have suggested holding conferences or other scientific processes to exchange views and practices that may help in successful online counseling.

“Exchange of experiences amongst individuals who work in different groups is necessary... participate in conferences and analyze case studies…” (P6).

“Education. It would help me to know the experience of others…. Networking around such topics”. (P3).

“Education is necessary...I recently discovered that there are tools out there that we are unaware of, like the life map”. (P16).

Additionally, several therapists offered solutions to technical and practical issues.

“The networks should be better and with optical fibers there are imperfections. There are some connection problems on certain days and times”. (P1).

“Good network that is open to everyone. Good know-how and digital accessibility. Encourage having a face-to-face first meeting. It will help in the therapeutic relationship”. (P3).

It has been suggested by a number of respondents that the network supporting online counseling be established within the framework of a local government-organized social policy and/or create specific platforms adapted to counseling.

“A strong network throughout the city would be useful. Consequently, local government could provide help. There are places, for example, where a fast network is not supported…” (P11).

“I think it would help to have a platform for such cases. Not skype or viber. To make it easier. It would help us therapists to promote such a platform. I have seen it in many colleagues who are promoting such platforms…” (P14).

## 4. Discussion

Our study’s findings demonstrate that some therapists had used online counseling occasionally, either as part of their training, or when a client had to leave the place of their treatment for a variety of reasons, such as employment or studies. However, the great majority of therapists started using online counseling more systematically during COVID-19. Despite their initial hesitation and the challenges they encountered, the majority of them reported that the overall experience was positive. Several therapists also had a positive experience with online counseling before the outbreak of the COVID-19 health crisis [9]. Most therapists both in our study and in others stated that when they started using online counseling during the pandemic, their previous experience or relevant training were very limited or completely absent [11].

The factors that, according to our research, can influence the course of a successful online consultation rely both on the organization itself and how well prepared it is, as well as on knowledge-related issues. Mainly, the therapists working in a public agency highlighted the role of the organization. This can be explained by the fact that private sector therapists operate with greater independence and have the resources, know-how, and training necessary to provide online therapy.

All interviewees put an emphasis on the need of having a distinct and clear contract for online counseling. This will eliminate many potential issues and outline a new treatment framework. Previous research has indicated that the existence of a protocol and client and therapist preparation are essential for the process of online counseling because, while it provides practical support, it also serves to lower stress and enhance the therapeutic process as a whole [9].

Our research suggests that the therapist has a crucial role. Their experience and prior knowledge and familiarity as well as the way in which they manage the issues and context of the treatment affect its quality [12,31]. Other findings also support the pivotal role and importance of the therapist’s experience in online counseling [7,31]. A therapist who does not feel comfortable with the internet will not be able to convey a feeling of safety to their clients to accept it [14].

In addition, the therapists’ psychotherapeutic approach influences the effectiveness of online counseling. Some professionals consider that there are limitations to some psychotherapeutic approaches including psychodynamics. Compared to their counterparts from other methods, such as psychoanalysts, cognitive approach psychotherapists prefer online counseling [15]. In the present study, systemic approach therapists accounted for the majority of participants who indicated a preference for online counseling. One interpretation of these data could be that the therapists of this approach are flexible enough to adopt techniques from other approaches and are receptive to any “innovation” in the field of psychotherapy.

The mode and mechanisms of change used in therapy may explain why some psychotherapeutic approaches favor online counseling. Both our study and others [9,15] report difficulties in the psychodynamic approach of therapists for online counseling. A possible interpretation is that the limited use of non-verbal communication in this particular technique and the difficulty of implementing the change mechanisms that it uses, such as the method of free association on the couch, do not leave room for its successful use [15].

A first evaluation of online counseling, according to our results, states that among its advantages are the arrangement of practical issues such as the reductions in distance, money, and travel time. In addition, the elimination of social exclusion for vulnerable people is an important contribution of online counseling. Related studies [14,32,33,34] have produced similar results. Of course, online counseling creates another kind of exclusion for those population groups without access to technology. People who belong to vulnerable groups who usually lag behind in terms of the knowledge and management of social media, or who lack an internet connection or the necessary logistical equipment, are thus excluded.

Our findings suggest that, despite the positives of online counseling, it is not recommended for specific categories of psychological disorders, such as interventions for people with psychosis or mental retardation. There are studies that positively evaluate the intervention of online counseling in specific disorders including anxiety and depression [35]. However, other research data highlight its weaknesses in specific disorders as well as in crisis situations, such as suicide attempts [17,18].

The participants of the present study emphasized repeatedly that online counseling does not allow the utilization of non-verbal communication or the use of certain therapeutic tools. This poses significant challenges for the course of the treatment because no information is gathered that could aid in its advancement and development, such as the therapeutic process or the analysis of the clinical picture, which would have provided the therapist with the necessary information to intervene, particularly in crisis situations [12].

According to some study participants, online counseling does not safeguard ethical issues like privacy. Other research has produced similar findings [14,33,35,36,37]. Mendes-Santos et al. [14] suggest that one plausible reason for the therapists’ inability to feel secure in the internet traps could be their inadequate training and lack of knowledge.

Consistent with other studies, divergent opinions on establishing a therapeutic relationship online were found in our results. Some believe that the development of a therapeutic relationship is hampered or prevented by distance [14,34,36,37,38]. Nonetheless, other therapists disagree, arguing that internet counseling fosters a positive therapeutic alliance [7].

The views of the participants presented in our study appear to recognize that online counseling is a practice that may spread rapidly in the future. This kind of practice may potentially assume a supplementary role in in-person counseling, in cases, for instance, of the clients’ disability or health issues that do not allow for physical attendance. Our findings point to the possibility of a uniform, reliable protocol for online therapy. As a result, a stable framework is established, facilitating communication between the therapist and the client as well as the advancement of the healing process. These perspectives corroborate findings from earlier research [7,8,9].

Finally, the sample’s participants endorse the need for therapists to have knowledge and training about online counseling. Knowledge and previous experience in using the internet increase the likelihood of its correct use in the future [39]. According to the APA guidelines for online counseling, therapists must make sure that they have adequate knowledge of the technology required and that online counseling is beneficial to their clients [11]. This is even more evident in the post-COVID-19 era in that online counseling is expected to have an increased utility for people who face health issues requiring systematic monitoring, for elderly people, and for people who experience all kinds of exclusion due to gender, religion, or spiritual peculiarities [40]. The readiness of the therapist to use online counseling should also be underlined. To effectively use these emerging technologies, the therapist should have appropriate skills and experience in managing sensitive issues such as the difficulties in applying online counseling in cases of domestic violence, couple therapy, or when the whole family should be present in the process [41]. The development of supportive technological environments and appropriate and accessible infrastructures are necessary to confine the reluctances of therapists and clients to use online practices within the context of a contemporary therapeutic process [9,41].

## 5. Limitations

In our study, a qualitative research method was used that does not allow for the generalization of our results. While our analysis covers therapists across Greece, it mostly focused on those in the greater Attica and Crete regions, with a few outliers. This has the consequence of limiting the viewpoints of therapists from smaller towns or other regions of Greece who may have had different perspectives. Moreover, not all therapist backgrounds are represented in our study sample; psychiatrists, for example, would likely have different perspectives and be able to offer us a broader understanding of the online counseling issue. In addition, the majority of the therapists in our sample employed a systemic psychotherapy approach. It is unknown if using different types of treatment methods would add new data to our research. Furthermore, the present study did not include any male therapists. Our findings thus demonstrate a feminine perspective of the topic under investigation. Although, this targeted perspective of our findings can be viewed as an added value for our study, readers should also consider that the gender of a therapist has an impact on their counseling practice. In addition, psychotherapy often investigates the articulation of emotions and the discourse of emotional matters in a manner that is more in line with the gender traits of the counseling therapist and diverges in terms of their communication style with the patient [42].

Finally, the period within which our survey was conducted inevitably affected our results. The therapists’ opinions of online counseling might have changed if it had been conducted at a different time, such as at the start or end of the pandemic. As such, the findings of this study should be viewed under these limitations.

## 6. Conclusions

To our knowledge, this study is one of the few attempts at reflecting on the working experiences of mental health counselors during the COVID-19 pandemic. It provides an idea about the adaptation of these professionals to the pandemic in Greece through the use of online counseling and may contribute to its appropriate use in the future.

Our findings indicate that online therapy has a number of benefits and can be utilized either exclusively or in conjunction with counseling in crisis situations like COVID-19. But future use of it necessitates preparedness. Therapists need to specialize in internet therapy and receive rigorous training in this area. It is still necessary to develop and enforce a treatment protocol that is consistent for all online counseling providers and adhered to by all parties involved.

According to our results, online counseling has several advantages and can be used in combination with counseling or entirely in crisis situations such as COVID-19. However, its use in the future requires preparation. In particular, the systematic training and specialization of therapists in online counseling is required. There is still a need to formulate and have a specific treatment protocol followed by all involved, which is uniform for all therapists practicing online counseling.

Both in the public and private sectors, there must be synchronization and coordination between those responsible for solving respective issues. The contribution of social policy at local and national levels is imperative. It will address training issues, and also finance the actions that need to be performed.

The present research can be continued in the future with other groups of therapists who practice online counseling, such as psychiatrists. It can be extended to other regions of Greece with different social and geographical conditions or combined with quantitative studies. Finally, it would be important to conduct an additional study that would present the views of the clients and their families regarding their experience with online counseling and providing their own suggestions for the future.

## Data Availability

Data generated during the present study cannot be shared due to issues with the subjects’ privacy and confidentiality.

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
