# Peer review of "Evaluation of Online Counseling through the Working Experiences of Mental Health Therapists Amidst the COVID-19 Pandemic"

_healthcare, 2024, doi:10.3390/healthcare12040495_

Round 1

Reviewer 1 Report

Comments and Suggestions for Authors

The manuscript by Moudatsou used a qualitative approach to evaluate 17 mental health professionals’ experiences with online counseling during the COVID-19 pandemic in Greece. Overall the manuscript is well-written and this is an interesting study. I have a few comments that I think could help strengthen the presentation of the methods and results.

  • in 2.3 Population and study sample, please provide more details on the recruitment process of mental health professionals. Please provide details on the sample population of your study. What’s your study’s sample population? In which country, area, city, and hospitals? Are there any inclusion and exclusion criteria? Seventeen out of how many professionals did you reach out for consent? How long was the recruitment period?
  • What’s the proportion of female mental health therapists in Greece? Since all the participants in your study are females, I think it limits the generalizability of your study findings. Please include one or two sentences in your study discussion to describe this potential limitation.

Reviewer 2 Report

Comments and Suggestions for Authors

Comments and Suggestions for Authors

It is an attempt on reflection for online counseling during the Covid19 pandemic among mental health professionals' experiences. Thanks for the opportunity to peer review.

Introduction

It would be interesting if the authors could refer to the critical need to find how the existing technologies have been appropriated across a range of online practices and to understand system requirements that might be more specific to creating effective platforms for therapists.

Methods

2.3 Population and study sample

1.Could authors describe detail technique about “The participants of the sample were selected via purposeful sampling technique.”?

2.What are the study criteria those individuals who meet?

2.4

How the authors decided the amount of sample size?

2.5

Could authors describe about what are the questions of the semi-structured interview consisted of?

2.6 data analysis

1.Please provide description the COREQ guidelines about 32-item checklist that the study results were used for reporting.

2.How authors ensured the findings were representative of participants' experiences and did not leave out any topic that was especially important to discuss.

3.Results

Author should corrected 3.1.1 to 3.2.1 at line 288.  

4.Discussion

Could the author describe more situation about “Everyone who participated in our survey feels that in the future, internet counseling should be used to supplement in-person counseling.” at lines 487.

Reviewer 3 Report

Comments and Suggestions for Authors

Dear authors, I read the article entitled “Evaluation of online counseling through the working experi- 2 ences of mental health therapists in the midst of the COVID-19 3 pandemic” with great interest. Readers of "MDPI Healthcare" might profit from your article. Overall, the manuscript addresses the current and critical issue of online counseling during the COVID-19 pandemic, offering insights into the experiences of mental health therapists. Moreover, the use of qualitative research with semi-structured interviews provides an in-depth exploration of therapists' perspectives. Additionally, involving mental health professionals from both public and private sectors enriches the study with varied experiences. However, I have some suggestions for the revision of the manuscript according to the standards of the MDPI and I encourage you to re-submit your article:

Introduction: Readers might profit from strengthening the introduction by clearly outlining the research gap your study aims to fill and its significance in the current context.

Participant Selection: Please consider discussing potential selection biases and how they might impact the results. In this term, addressing the homogeneity of the sample (e.g., all participants being women) might be an crucial add-on.

Data Analysis: Providing more details on the framework analysis process, including how themes were derived and any measures taken to ensure the validity and reliability of the findings, might enrich the methods section. Moreover, please consider discussing how these findings can inform future practices in online counseling, particularly in crisis situations like a pandemic.

Discussion of Findings: Readers might profit from Providing an enhanced discussion by comparing findings with existing literature more robustly, particularly focusing on the implications for online counseling practices post-pandemic. In addition, you may include a more comprehensive analysis of how online counseling can be integrated into routine mental health services post-pandemic.

By addressing these areas, the manuscript can make a significant contribution to the field of healthcare, particularly in understanding and improving online counseling practices, aligning with the standards and expectations of MDPI Healthcare.

Comments on the Quality of English Language

Proofreading the manuscript for language clarity and grammatical accuracy might be a profit.

Round 2

Reviewer 3 Report

Comments and Suggestions for Authors

I confirme that the revised paper satisfies my comments. So, I do not have any more comments for the authors.

Comments on the Quality of English Language

Minor editing of English language required.